# SFRP2 Overexpression Induces an Osteoblast-like Phenotype in Prostate Cancer Cells

**DOI:** 10.3390/cells11244081

**Published:** 2022-12-16

**Authors:** Elif Akova Ölken, Attila Aszodi, Hanna Taipaleenmäki, Hiroaki Saito, Veronika Schönitzer, Michael Chaloupka, Maria Apfelbeck, Wolfgang Böcker, Maximilian Michael Saller

**Affiliations:** 1Department of Orthopaedics and Trauma Surgery, Musculoskeletal University Center Munich (MUM), Ludwig-Maximilians-University (LMU) Hospital, Fraunhoferstraße 20, 82152 Planegg-Martinsried, Germany; 2Institute of Musculoskeletal Medicine (IMM), Musculoskeletal University Center Munich (MUM), LMU Hospital, Fraunhoferstraße 20, 82152 Planegg-Martinsried, Germany; 3Urologischen Klinik und Poliklinik, LMU Hospital, Marchioninistr 15, 81377 München, Germany

**Keywords:** SFRP2, osteomimicry, prostate cancer, PC3, bone metastasis, EMT, WNT signaling

## Abstract

Prostate cancer bone metastasis is still one of the most fatal cancer diagnoses for men. Survival of the circulating prostate tumor cells and their adaptation strategy to survive in the bone niche is the key point to determining metastasis in early cancer stages. The promoter of *SFRP2* gene, encoding a WNT signaling modulator, is hypermethylated in many cancer types including prostate cancer. Moreover, SFRP2 can positively regulate osteogenic differentiation in vitro and in vivo. Here, we showed SFRP2 overexpression in the prostate cancer cell line PC3 induces an epithelial mesenchymal transition (EMT), increases the attachment, and modifies the transcriptome towards an osteoblast-like phenotype (osteomimicry) in a collagen 1-dependent manner. Our data reflect a novel molecular mechanism concerning how metastasizing prostate cancer cells might increase their chance to survive within bone tissue.

## 1. Introduction

Prostate cancer is the second most abundant cancer type in males, and even with the high treatment success after early diagnosis, it is still the fifth cancer-related death cause worldwide [1]. In advanced prostate cancer (AdPCA), when cancer cells metastasize outside the primary location, the survival rate decreases dramatically due to currently unavailable therapeutic options and resistance to androgen depletion [2].

The most abundant metastatic site of AdPCA is bone, especially the spine, hip, and long bones [3]. Other cancer types, such as lung, breast, and kidney, also have a high potential to metastasize to bone at an advanced stage, but, unlike prostate cancer, these cancers cause bone resorption, whereas AdPCA can result in both bone-forming (osteoblastic) and bone-resorbing (osteolytic) metastasis [4]. However, the molecular pathways that underly prostate cancer bone metastasis and osteoblastic or osteolytic progression are still poorly elucidated.

Bone metastasis can be separated into early and late phases that are defined by specific time-dependent molecular pathways and interaction with local cells. In early metastasis, just after the cancer cells entered into and settled in the bone, colonization, survival, dormancy, and reactivation occur until the late metastasis stage, when the tumor starts to actively grow [5]. Only around 0.01% of circulating cancer cells can survive and adapt to the bone niche during the early bone metastasis stage [6]. In the late metastatic stage after the formation of bone macrometastasis, the survival rate of cancer patients drops dramatically, by almost 60% [7]. Therapeutically, it is therefore incredibly important to diagnose and intervene in bone metastases already at the early metastasis stage.

The PC3 cell line, derived from a grade IV bone metastatic prostatic adenocarcinoma, is an aggressive metastatic prostate cancer cell line [8]. Injection of PC3 cells into the murine tibiae leads to bone lesions [9], while silencing *Dickkopf-1* (*DKK1*), an inhibitor of the WNT (Wingless/Int) signaling, was found to allow PC3 cells to secrete osteoblast-related factors and increase alkaline phosphatase activity in bone marrow stromal cells [10]. Furthermore, injection of a *DKK1*-overexpressing prostate cancer cell line C4-2B reduces bone mineral density and, therewith, leads to the formation of osteoblastic lesions in a murine animal model [11]. Overall, the currently available data suggest that WNT signaling activity is involved in the balance between osteolytic and osteoblastic phenotypes in prostate cancer bone metastasis.

WNTs are secreted glycoproteins that bind to various cell surface receptors to modulate many cellular activities such as proliferation, differentiation, apoptosis, migration, invasion, and tissue homeostasis [12]. WNT signaling cascades are divided into the canonical WNT/ß-catenin pathway that is activated via the interaction of WNT with Frizzled receptors (FZDs) and LRP5/6 (low-density lipoprotein receptor-related protein 5/6), and the ß-catenin independent non-canonical pathways, such as WNT/Ca^2+^ signaling and WNT/PCP (planar cell polarity) signaling [13]. Aberrant WNT signaling is associated with the development of bone metastasis in prostate cancer [14]. Similar to other common cancer types, WNT signaling is continuously active in prostate cancer due to a downregulation of inhibitory WNT regulator genes, such as *Wnt Inhibitory Factor 1* (*WIF1*), *Dickkopf proteins* (*DKK1, DKK2, DKK3*), and *Secreted Frizzled Related Proteins* (*SFRP1, SFRP2, SFRP3, SFRP4*), usually caused by methylation of the respective promoter [15,16]. WIF1 and SFRPs can directly bind to WNT ligands, which, in turn, leads to a WNT signaling inhibition [15]. Moreover, SFRPs and DKKs bind competitively to receptors that are involved in WNT signaling. While DKKs can bind to the co-receptors of LRP5/6, SFRPs can bind to FZDs [16]. Despite the autocrine effect of these WNT signaling inhibitors on primary cancer cells, their paracrine effect on bone cells during metastasis could help to understand the molecular pathways underlying osteolytic or osteoblastic bone metastasis.

SFRP2 is a soluble extracellular protein that acts as a WNT signaling modulator. Although SFRP2 is known to prevent the binding of WNT ligands to their respective receptors [17], recent research has shown that it augments WNT16B and WNT3A to promote WNT signaling in advanced malignancies [18] and HEK293 cells [19], respectively. The modulatory activity of SFRP2 is still unclear and, therefore, represents a potential target for understanding its role in cancer progression. It is known that the *SFRP2* promoter is hypermethylated in primary prostate cancer, and, thus, *SFRP2* expression is silenced relative to adjacent prostate tissues [20]. This suggests that SFRP2 may act as a tumor suppressor in many cancer types, but recent studies have shown that SFRP2 could also act as a tumor promoter, depending on the cell type or cellular location [21,22]. Moreover, elevated serum levels of SFRP2 are associated with poor prognosis and metastasis in breast cancer patients, in which the *SFRP2* promoter is, similar to prostate cancer, hypermethylated in the primary cancer side [23].

Additionally, SFRP2 is the only negative regulator of WNT signaling that contributes to both bone morphogenesis (Gene Ontology: GO:0060349) and osteoblast differentiation (GO:0001649), and, thus, SFRP2 could be a potential regulator of bone metastasis, similar to *DKK1*. *Sfrp2*-deficient mice show poor osteogenic differentiation, whereas administration of recombinant SFRP2 restores the expression of the osteogenic markers *Runx2* and *Osx* in vitro [24]. Likewise, due to its interaction with the fibronectin–integrin complex, SFRP2 can increase cell adhesion, which is an essential prerequisite for cancer cells to survive in the metastatic bone niche [25]. These findings suggest that SFRP2 expression may play a key role in the osteotropic activity of cancer cells after leaving the primary location.

Prostate cancer cells must substantially adapt their transcriptional landscape in the metastatic bone target area to maintain their viability. This so called osteomimicry is a survival strategy of cancer cells in the bone during early metastasis and is characterized by the secretion of bone-related proteins to cope against immune system reactions [26]. AdPCA cells acquire an osteoblast-like phenotype and secrete bone matrix proteins and osteoblast-related factors to mimic bone-related cells in the target niche [27]. After this dormancy step, depending on the secreted proteins, an increase or decrease in bone formation is induced. Therefore, SFRP2 promotes an osteoblastic phenotype in early metastasized prostate cancer cells and therewith promotes cell survival, or it additionally induces osteoblastic bone lesion in later stages due to an increase in osteoblastic differentiation. SFRP2 is a prospective target to detect poor prognosis of AdPCA.

Epithelial-mesenchymal transition (EMT) is another critical concept in the cellular plasticity of cancer cells during early metastasis. EMT allows cancer cells to survive during circulation until they reach their target tissue [28] by changing their morphology to a mesenchymal-like phenotype and altering biochemical pathways. The transformation of epithelial cancer cells into a mesenchymal cell phenotype increases their invasion potential into target tissues, including bone [29]. This fact underlines EMT as a potential marker for monitoring osteomimic features of cancer cells during early bone metastasis [30].

In this study, we found that overexpression of SFRP2 in the prostate cancer cell line PC3 increases the expression of osteoblast-like genes, when seeded on the most abundant bone extracellular matrix (ECM) structural protein, collagen 1 (COL1). Moreover, overexpression of SFRP2 alters the proliferation rate and morphology to a mesenchymal-like phenotype and increases the adhesion of PC3 cells when compared to a non-COL1-coated surface. These results emphasize that SFRP2 is a promising key element for prostate cancer cells to survive in the bone metastatic niche through osteomimicry during early bone metastasis.

## 2. Materials and Methods

### 2.1. Cell Line and Cell Culture Conditions

The PC3 cells, derived from bone metastasis of human prostate adenocarcinoma, were obtained from ATCC, USA. Cells were cultured for up to 10 passages in Dulbecco’s Modified Eagle Medium + GlutaMax (Thermo Fisher, Waltham, MA, USA), with 10% fetal bovine serum (FBS, Sigma Aldrich, St. Louis, MI, USA) and 1% penicillin/streptomycin (Sigma Aldrich, USA) at 37 °C with 5% CO_2_. The media were changed every 3 days. Due to the relatively low doubling time and the ability of cancer cells to grow on top of each other, the confluence was kept below 90% to avoid any disturbance of cell–surface interaction.

### 2.2. Generation of PC3 Cells Stably Overexpressing SFRP2

To understand the intrinsic and extrinsic effect of SFRP2 protein on prostate cancer cells, the coding sequence of the human *SFRP2* gene was stably expressed in PC3 cells by using a Sleeping Beauty transposon system. The Sleeping Beauty plasmid backbone pSBbi-RP (60513, Addgene, Watertown, MA, USA) [31] was modified in our laboratory by removing an EcoRI side with respective restriction enzymes and adding an extra multiple cloning sites (MCS) with the sequence for 6× His-Tag (HIS) (pSBbi-RP-deltaECO-HIS) by ligation. Initially, the human *SFRP2* coding sequence was digested from a pCMVSPORT6 vector (6423, Horizon Discovery, Waterbeach, UK) with EcoRI and EcoRV (NEB, Ipswich, MA, USA), purified with NucleoSpin Gel and PCR Cleanup kit after gel electrophoresis (MN, Düren, Germany), and inserted into the pSBbi-RP-deltaECO-HIS backbone with the same restriction enzymes. After overnight ligation of the insert to the backbone at 4 °C, plasmids were inserted into *E.coli* DH5α by a heat shock transformation at 42 °C, and positive transformants were expanded in LB medium with ampicillin (Thermo Fisher, USA) overnight. To generate SFRP2 overexpressing (PC3*^SFRP2^*) or control cells (PC3), 0.4 µg of the newly created plasmid, 0.9 µg transposase pCMV (CAT) T7-SB100 (24879, Addgene, Watertown, MA, USA), and 20 µL nucleofector solution (Amaxa™ Cell Line Nucleofector™ Kit V, Lonza, Basel, Switzerland) were added to 500,000 PC3 cells and electroporated with the 4D-Nucleofector™ Core Unit (program B 032, Lonza, Switzerland). Control PC3 cells were transfected with an empty pSBbi-RP-deltaECO-HIS backbone without the insert. After transfection, cells were treated with 2 µg/µL puromycin (Sigma Aldrich, St. Louis, MI, USA) for 14 days to kill non-stably transfected cells. For further purification, cells were sorted according to their RFP expressions by fluorescent-activated cell sorting (FACS). The size and granularity of the RFP expressing PC3 and PC3*^SFRP2^* cells were analyzed by forward and side scatter gating of the FACS results.

### 2.3. Surface Coating

Collagen 1 is the most abundant protein in the bone microenvironment and may, therefore, influence the response of prostate cancer cells. Therefore, we decided to mimic the bone microenvironment by coating the surface of all used cell culture-treated polystyrene (CCP) flasks or wells (Thermo Fisher, USA) and chambers (ibidi, Gräfelfing, Germany) with 20 µg/mL collagen 1, isolated from rat tails (Sigma-Aldrich, St. Louis, MI, USA) overnight at 4 °C and rinsing once with phosphate-buffered saline (PBS) before experiments. For all experiments, non-COL1-coated CCP flasks or wells served as controls and are referred to as CCP. In sum, we analyzed 4 different groups: CCP-PC3, CCP-PC3*^SFRP2^*, COL1-PC3, and COL1-PC3*^SFRP2^*.

### 2.4. Validation of SFPR2 Overexpression

To validate the overexpression of *SFRP2* at the transcriptional level, PC3 and PC3*^SFRP2^* cells were seeded at a density of 8000 cells/cm^2^ in COL1-coated and CCP T25 flasks. After the cells reached 90% confluency in about 3–4 days, total RNA was isolated with TRIzol (Invitrogen, Waltham, MA, USA) and purified with a Direct-Zol RNA Kit (Zymo Research, Irvine, CA, USA). cDNA was synthesized on PEQSTAR (PEQLAB, VWR, Radnor, PA, USA) using the Biozym cDNA Synthesis Kit (Biozym, Hessisch Oldendorf, Germany), with 1 µg of RNA and random hexamer primers, according to the manufacturer’s protocol. qPCR was performed with specific FAM-labeled *SFRP2* primers (IDT, Coralville, USA) and Promega GoTaq qPCR kit (Promega, Madison, WI, USA), according to the manufacturer’s protocol, with a LC96 LightCycler (Roche Applied Sciences, Penzberg, Germany). Human *HPRT* (IDT, Coralville, IA, USA) was used as a reference.

To validate the protein level of SFRP2 in PC3 cells, 15,000 cells/cm^2^ were seeded in 8-well chamber slides (CCP: 8-well high µ-Slides, COL1: 8-well glass µ-Slides; both from ibidi, Gräfelfing, Germany). After 24 h, attached cells were rinsed with PBS, fixed with 4% PFA in PBS (Merck, Darmstadt, Germany) for 10 min at room temperature, and permeabilized with 1% Triton-X-100 (Sigma, USA). Immunocytochemistry (ICC) against SFRP2 was performed with a polyclonal rabbit@SFPR2 antibody (ab137560, Abcam, Cambridge, UK) and a goat@secondary antibody with conjugated Alexa Fluor 488 (ab150113, Abcam, Cambridge, MA, USA). Images were taken with an inverted epifluorescence microscope (AxioObserver, Zeiss, Jena, Germany).

### 2.5. Cell Proliferation, Metabolic Activity and Morphology

To analyze the consequence of SFRP2 overexpression on the proliferation of PC3 cells on COL1-coated and CCP surfaces, 2000 cells/cm^2^ were seeded in T25 flasks and counted every 3–4 days with a hemocytometer. After counting, 2000 cells/cm^2^ were re-seeded to the same flask, and the seed-count-cycle was repeated for up to 15 days. The population doubling time (*PDT*) and cumulative population doublings (*cumPD*) were calculated using the following formulas:$$PDT\left(\bar{x}\;for\;days\right)=\frac{\left(Number\;of\;days\;passed\;between\;two\;passages\;x\;24\right)\;x\;\log\left(2\right)}{\log\left(Number\;of\;counted\;cells\right)-\log\left(Initial\;number\;of\;cells\;seeded\right)}$$
$$cumPD\;\left(\bar{x}\;for\;days\right)=\frac{\ln\frac{\left(Number\;of\;counted\;cells\right)}{\left(Initial\;number\;of\;cells\;seeded\right)}}{\ln\left(2\right)}$$

Before each passaging, images of each experimental group were taken to analyze morphological differences. At least 100 cells per passage were analyzed for cell area and cell shape with ImageJ [32]. The cutoff for both area and aspect ratio is defined as 3 for better visualization in violin plots.

To analyze SFRP2-dependent changes in the metabolic activity of PC3 cells on CCP and COL1 surfaces, a WST-1 colorimetric assay was performed. Therefore, 10,000 cells/well were cultured overnight and afterwards incubated with the WST-1 Reagent (Roche Applied Sciences, Penzberg, Germany) in 1:10 final dilution in complete media for 4 h. The colorimetric change was measured on a Multiskan FC microplate reader (Thermo Fisher, USA) at 450 nm for formazan and 620 nm as reference.

To test the efficiency of a single PC3 or PC3*^SFRP2^* cell to form colonies on CCP or COL1 surfaces, 400 cells/well were seeded in 6-well plates. After 10 days, newly formed colonies were fixed with 4% PFA in PBS and stained with 0.1% Crystal Violet in isopropanol (Sigma-Adrich, St. Louis, MI, USA). Colonies with at least 20 cells were counted, and colony forming unit (CFU) efficiency was calculated and normalized to CCP-PC3 cells.

### 2.6. Migration, Invasion and Attachment

To investigate if there is a relation between SFRP2 overexpression in PC3 cells and COL1 interaction, we analyzed the migration, invasion, and attachment characteristics. Random migration of PC3 and PC3*^SFRP2^* on CCP and COL1 surfaces were observed with a time lapse microscope (Axiovert S100, Zeiss, Jena, Germany) at 37 °C 5% CO_2_. Therefore, 5000 cells/well were seeded in 6-well plates, incubated overnight, and imaged every 15 min for 48 h. At least 100 cells per experiment were analyzed utilizing ImageJ and MTrackJ [33].

The invasion potential of PC3 or PC3*^SFRP2^* through COL1-coated membranes was analyzed by a transwell assay with a membrane pore size of 12 µm (3403, Corning, Corning, NY, USA). Therefore, 50,000 cells/insert were seeded in media without FBS to the upper compartment. Media with 10% FBS in the lower compartment served as chemoattractants. After 6 h of incubation, cells were fixed with 4% PFA in PBS and stained with 0.01% DAPI in PBS. Images were taken on an inverted epifluorescence microscope (AxioObserver, Zeiss, Germany), and cells were automatically counted with the “find maxima” function of ImageJ.

To test the effect of SFRP2 overexpression on cell adhesion, 10,000 cells/well were seeded on COL1-coated or uncoated 96-well plates in serum-free medium and washed after 2 h of incubation with PBS. As a negative control, 1% BSA in PBS was used. The remaining adhered cells were stained with 0.1% crystal violet in methanol. The colorimetric measurement was performed on Multiskan FC Microplate Photometer (Thermo Fischer, Waltham, USA) at 570 nm.

### 2.7. Next Generation RNA Sequencing and Bioinformatics Analysis

To identify SFRP2-induced transcriptional changes in PC3 cells, 100,000 cells were seeded in T25 flasks, and RNA was isolated after 5 days, as described in Section 2.3., and RNA integrity was validated with Bioanalyzer (Agilent, Santa Clara, CA, USA). cDNA sequencing libraries were prepared with a SENSE mRNA-Seq Library Prep Kit V2 (Lexogen, Vienna, Austria). Sequencing was performed on a HiSeq1500 device (Illumina, San Diego, CA, USA) with a read length of 50 bp and a sequencing depth of approximately 12 million reads per sample.

Obtained FASTQ files were demultiplexed with sample-specific barcodes, and adaptor sequences were clipped before passing to the alignment. Reads were aligned to the homo sapiens reference genome (release GRCh38.101) using STAR (version 2.7.2b) [34].

Genes with less than 10 reads in all samples were filtered out, leaving 22,105 genes for further analysis. Gene expression was normalized with a variance stabilizing transformation (vst), which was further utilized for a principal component analysis (PCA).

Differentially expressed genes were calculated with the DESeq2 package (version 1.28.1) [35] in R (version 4.0.3). Significantly changed genes were determined by utilizing an adjusted *p*-value (*p*-adj) of <0.05 and a Log2FoldChange of ±2 as cut-offs and utilized for the MA-Plot. Gene set enrichment analysis (GSEA), utilizing significant changed genes without a Log2FoldChange cut-off (COL1-PC3 vs. COL1-PC3*^SFRP2^*: 7174 significantly changed genes; CCP-PC3 vs. CCP-PC3*^SFRP2^*: 88 significantly changed genes), was performed separately for each group with the clusterProfiler package (version 4.2.2) [36] to understand the match of significantly upregulated genes in each surface condition with gene ontology biological pathways.

By using the GO:0001649 osteoblast differentiation pathway gene set and our DESeq2 results without Log2FoldChange cut-offs, a matched gene set was created. Out of 158 genes in GO:0001649, 46 genes were found to be differentially expressed in between COL1-PC3 and COL1-PC3*^SFRP2^*. These genes were used to create a heatmap and clustering analysis with normalized gene expressions, including all groups.

### 2.8. Statistical Analysis

All experiments were performed in triplicates and repeated at least three times. Statistical calculations were performed in R-Studio (version 4.1.0). After the verification of the Gaussian distribution and variance of the results, a two-way ANOVA was performed for two independent variables and their interactions. Significance of both the effect of surface coating and cell type and their interactions were calculated separately. Depending on the F-stat significance, multiple pairwise-comparison was performed with TukeyHSD (Tukey honest significant differences). Statistical significance was defined at a *p*-value of <0.05.

## 3. Results

### 3.1. Generation and Validation of PC3 Cells Stably Overexpressing SFRP2

As the *SFRP2* promoter is hypermethylated in many cancer cells, including PC3 cells, we created PC3 cells that stably overexpress SFRP2 by utilizing the Sleeping Beauty transposon system [37]. The unmodified bicistronic expression plasmid with coding sequences for red fluorescence protein (RFP/dTomato) and puromycin resistance was used for establishing stably transfected control cells (PC3), while the same plasmid with an insertion of the human coding sequence of human *SFRP2* was used to generate the *SFRP2* overexpressing cells (PC3*^SFRP2^*, Figure 1A). After 14 days of puromycin selection, most cells stably expressed RFP (Figure 1B). To further enrich PC3 cells expressing high RFP and hence high *SFRP2*, FACS was performed (Figure 1B′), resulting in homogenous, bright RFP fluorescent cell populations (Figure 1B″).

To validate *SFRP2* overexpression on transcriptional and translational levels, we performed qPCR and immunocytochemistry (ICC) for PC3*^SFRP2^* and PC3 cells seeded on COL1-coated and CCP surfaces. qPCR showed that *SFRP2* expression is more than 20-log2fold higher in PC3*^SFRP2^* cells in comparison to control PC3 cells, independent of the cell culture surface (Figure 1C). Moreover, ICC revealed a clearly higher expression of SFRP2 protein in PC3*^SFRP2^* cells in comparison to PC3 (Figure 1D,D′). In addition, these results specifically demonstrated that SFRP2 expression was not affected by different culturing methods in PC3*^SFRP2^* cells.

### 3.2. SFRP2 Overexpression Promotes a Higher Number of Differentially Expressed Genes in PC3 Cells on the COL1-Coated Surface Compared to the CCP Surface

To analyze the cell-intrinsic effect of SFRP2 overexpression in PC3 cells on both surfaces, we performed next generation RNA-sequencing. Interestingly, while just 17 genes were significantly differentially expressed in PC3 and PC3*^SFRP2^* cells, when cultured on the CCP surface (Figure 2A, Appendix A), culturing the cells on COL1 led to a significant increase of 453 differentially expressed genes (Figure 2A′). On the CCP surface, analysis of the gene ontology-related biological process of the upregulated genes in PC3*^SFRP2^* was related to ER stress (Appendix A). Yet, the downregulation of the type II collagen gene (*COL2A1*) in PC3*^SFRP2^*, which is involved in cancer stemness in various cancer types [38], and the upregulation of *CDH15* (Cadherin 15) in PC3*^SFRP2^*, which mediates intracellular adhesion [39], are promising predictors of the role of SFRP2 in cancer metastasis. When we compared gene expressions of PC3 and PC3*^SFRP2^* cells that were seeded on COL1, PC3*^SFRP2^* cells showed a significant COL1-dependent transcriptional adaptation, with 453 significantly differentially expressed genes (DEGs) (Figure 2A′, Appendix A).

When we investigate the upregulated genes in COL1-PC3*^SFRP2^* in detail, we discovered several genes that might contribute to the formation of osteomimicry. Members of the integrin (ITG) and matrix metalloproteinase (MMP) families mediate cell adhesion and invasion on or through extracellular matrices, respectively [40]. Interestingly, *ITGB2 (Integrin Subunit Beta 2*), *MMP11 (Matrix Metallopeptidase 11)*, and *MMP17 (Matrix Metallopeptidase 11)* were significantly upregulated in COL1-PC3*^SFRP2^* cells when compared to COL1-PC3. Moreover, increased expression of *TNXB* (*Tenascin XB*) causes an increase in cell adhesion [41], and *VWF* (*Von Willebrand Factor*) is a major platelet adhesion ligand [42] that contributes to the adhesion property of COL1-PC3*^SFRP2^* cells. Similarly, upregulation of *PTN* (*Pleiotrophin/Osteoblast-Stimulating Factor 1*) and *SPARC* (*Secreted Protein Acidic and Cysteine Rich*), which represent biochemical markers for osteoblast-like cells [43,44], were also detected in COL1-PC3*^SFRP2^* cells (Figure 2A′, Appendix A).

As SFRP2 is a WNT signaling inhibitor, we specifically investigated the autocrine effect of SFRP2 on PC3 cells by WNT signaling-related gene set (GO:0016055). Interestingly, we did not observe a significant effect of SFRP2 overexpression on WNT signaling in PC3 when the cells were cultured on CCP surface (Appendix A). *CTNNB1* encoding ß-catenin, which is an essential marker of canonical WNT signaling, was differentially expressed between PC3 and PC3*^SFRP2^* on COL1-coated surface, but the Log2FoldChange was below the cut-off value ± ∣2∣ (Appendix A). On the other hand, we found that WNT signaling-related genes such as *WNT4* and *WNT10B*, which are potential promoters of osteogenic differentiation [45,46], were significantly upregulated in COL1-PC3*^SFRP2^* in comparison to COL1-PC3 (Figure 2A′, Appendix A). In summary, differential gene expression analysis genes indicated the upregulation of some WNT signaling-related genes responsible in the osteogenic lineage in COL1-PC3*^SFRP2^* cells when compared to COL1-PC3 cells.

PC3 cells cultured on CCP surface showed almost no COL1-dependent transcriptional adaptation when compared to PC3 on COL1 (Appendix A). In contrast, the expression of 451 genes were significantly changed in COL1-PC3*^SFRP2^* cells in comparison to CCP-PC3*^SFRP2^* cells (Appendix A).

Furthermore, by directly comparing DEGs in all groups using a Venn diagram, we observed that COL1-PC3*^SFRP2^* cells have the highest number of significantly upregulated (Figure 2B) and downregulated (Appendix A) genes when compared to all other groups. In addition, a principal component analysis (PCA) also showed that the surface induced only negligible transcriptional changes in PC3 cells, in contrast to PC3*^SFRP2^* cells that were cultured on COL1, as COL1-PC3*^SFRP2^* showed a clear shift of the PCA cluster in comparison to CCP-PC3*^SFRP2^*.

### 3.3. SFRP2 Overexpression Leads to Osteotropic-like PC3 Cells on the COL1 Surface

To analyze the intrinsic effect of the 453 significant DEGs on the COL1 surface due to SFRP2 overexpression (Appendix A), we first performed a gene ontology (GO) enrichment analysis to detect the corresponding biological process that may contribute to the metastasis process (Figure 3A). When we examined the top 20 significantly upregulated biological process in COL1-PC3 and COL1-PC3*^SFRP2^*, we found that *SFRP2* overexpression causes a significant upregulation of genes that are associated with “Cell morphogenesis involved in differentiation”, indicating a possible EMT required for cancer metastasis, as well as with “Biological Adhesion” and “Cell Adhesion”, which may be required for osteotropic relocation of metastatic prostate cancer cells. The increase in “Taxis”- and “Locomotion”-related gene expression also supports the hypothesis that SFRP2 is involved in the relocation property of PC3 cells.

### 3.4. SFRP2 Overexpression Alters the Transcriptome of PC3 Cells towards an Osteoblast-like Phenotype on the COL1 Surface

Osteomimicry of metastatic cancer cells is defined by a molecular and phenotypic transition towards osteoblast-like cells in the bone environment. Therefore, we specifically evaluated genes that are involved in osteoblast differentiation (GO:0001649) within each experimental group. Interestingly, our analysis showed that, similar to the PCA (Figure 2C), COL1-PC3*^SFRP2^* is characterized by a unique cluster of osteoblast differentiation-related genes, when compared to all other groups (Figure 3B). Specifically, even though SFRP2 is accepted as a negative regulator of WNT signaling [47], SFRP2 overexpression leads to a COL1-dependent increase in *WNT4* and *WNT10B,* as well as *CTNNB1* (ß-catenin) expression (Figure 3B). Besides the specific WNT proteins that can promote osteoblast differentiation, COL1-PC3*^SFRP2^* cells also showed an increased expression of *BMP4* in comparison to all other groups*,* which is one of the most important osteoblastic lineage markers [48]. It is also known that prostate cancer cells secreting higher levels of BMP4 exhibit enhanced bone formation in vivo [49]. In addition, downregulation of the BMP receptors *BMPR1A*, *BMPR1B*, and *BMPPR2* in COL1-PC3*^SFRP2^* suggests that the secreted BMPs might be not utilized by the cancer cells in an autocrine manner.

### 3.5. SFRP2 Overexpression Induces COL1-Dependent EMT in PC3 Cells

“Cell morphogenesis involved in differentiation” (GO:0010770) is defined as a change in cell size and shape due to the differentiation process [50,51]. EMT is a crucial morphology-altering step for all primary cancer cells in early metastasis [52]. As our GO analysis results indicated that SFRP2 significantly increases the expression of cell morphogenesis-related genes in PC3*^SFRP2^* cells cultured on a COL1 surface (Figure 3A), we evaluated SFRP2-dependent changes of cell area and aspect ratio of PC3 and PC3*^SFRP2^* cells on both surfaces in vitro.

PC3 and PC3*^SFRP2^* cells did not show any differences in volume (Figure 4A, FSC-A) or granularity (Figure 4A, SSC-A) in suspension. However, while PC3 cells that were cultured on a CCP surface revealed the typical epithelial morphology, CCP-PC3*^SFRP2^* showed a more roundish morphology (Figure 4B). In contrast, we observed that both cell types adopted an elongated cell morphology and increased cell area when cultured on COL1 (Figure 4B′). COL1 surface significantly increases the cell area in both cell types in a SFRP2-independent manner (Figure 4C). Intriguingly, COL1-PC3*^SFRP2^* displayed significant cell elongation when compared to COL1-PC3 controls (Figure 4D). Furthermore, culturing PC3*^SFRP2^* cell on COL1 led to a significant increase in cellular elongation in comparison to CCP-PC3*^SFRP2^*.

This significant increase in the aspect ratio of PC3*^SFRP2^* cells in comparison to native PC3 cells suggested that phenotypical EMT only occurs on the COL1 surface in a SFRP2-dependent manner (Figure 4D). To strengthen this finding on a transcriptional level, we matched the genes that are significantly upregulated in COL1-PC3*^SFRP2^* and CCP-PC3*^SFRP2^* cells with hallmark gene sets for EMT (Figure 4E and Appendix A) and identified that 29 genes were upregulated related to Hallmark: EMT in COL1-PC3*^SFRP2^*, including *FBLN1, FBLN2,* and *TGFB1*, when compared to CCP-PC3*^SFRP2^* (Appendix A). Taken together, high expression of SFRP2 in PC3 cells positively regulates essential EMT-related genes and therewith promotes cell elongation in a COL1-dependent manner.

### 3.6. SFRP2 Overexpression Impedes the Reduction of Proliferation of PC3 Cells on COL1

Besides EMT, increased proliferation and metabolism inside the bone metastatic site, without being recognized by immune cells, is an additional crucial adaptation of cancer cells in early metastasis [53,54]. Therefore, proliferation was assessed by a cumulative population doubling, as well as a colony formation (CFU) assay.

Interestingly, while COL1-PC3 cells proliferate significantly slower, when compared to CCP-PC3 cells, the proliferation of PC3*^SFRP2^* cells was not changed when seeded on COL1 (Figure 5A,B).

Similarly, the COL1 surface significantly reduced the CFU efficiency in each group when compared to CCP-PC3 (Figure 5C). Interestingly, the lowest CFU efficiency was observed in COL1-PC3*^SFRP2^*, which also supported potential EMT on COL1, as EMT causes loss of intercellular adhesion [55].

Moreover, the lower CFU efficiency, but similar proliferation rate between PC3 and PC3*^SFRP2^* in each condition, was accompanied with a strong reduction of cellular metabolism in NCCPunc-PC3*^SFRP2^* cells compared to CCP-PC3. However, the difference was no longer significant when cells were cultured on the COL1 surface.

### 3.7. SFRP2 Overexpression Enhances COL1-Dependent Migration, Invasion, and Metabolic Activity of PC3 Cells

After cancer cells leave the primary site, their biochemical adaption to migrate, invade, and adhere to the target side are crucial to complete early metastasis. The secreted proteins might help us to define the migration of cancer cells and the location of metastasis [56]. Our GO pathways analysis (Figure 3A) shows that SFRP2 overexpression enhances cell adhesion, taxis, and locomotion of PC3 cells when cultured on COL1 surface. These findings suggest that SFRP2 may contribute to the osteotropism of metastatic prostate cancer cells. To validate our bioinformatics analysis, we investigated random migration, invasion through a COL1 matrix, and adherence of PC3 and PC3*^SFRP2^* cells on COL1 and CCP surfaces.

Interestingly, random migration assay demonstrated that the velocity of PC3 and PC3*^SFRP2^* cells was not significantly different on the same surface, but COL1-PC3 cells had a significantly faster random migration when compared to CCP surface (Figure 6A). In contrast, when cells had to migrate through a COL1 matrix, we counted significantly more PC3*^SFRP2^* cells in the lower compartment in comparison to PC3 cells (Figure 6B). Evaluation of cell attachment revealed that significantly more PC3*^SFRP2^* cells adhered after 2 h on CCP when compared to CCP-PC3. However, this effect was not significant between COL1-PC3 and COL1-PC3*^SFRP2^* (Figure 6C).

## 4. Discussion

This manuscript attempts to provide new information on mechanisms in the process of prostate cancer bone metastasis. After prostate cancer cells leave the primary site, demethylation of the *SFRP2* promoter likely drives cancer cells towards bone to eventually metastasize. Osteomimicry is a key element of cancer cell survival during early metastasis [57], and SFRP2 may be one of the crucial molecular players that modulates cancer cells and their environment to increase their adaptation to evade the immune response (Figure 7). In this study, we aimed to find a potential solution to intervene in the process of metastasis before aggressive late metastasis, when the balance of bone formation and resorption is disturbed, and additional biochemical mechanisms are involved [58]. Evasion of osteomimicry property of cancer cells by inhibiting SFRP2 is a possible mechanism to suppress bone metastasis. Moreover, SFRP2 might be a potential molecular candidate to detect bone metastasis in prostate cancer patients before fatal outcomes develop.

Our observations suggested that SFRP2 alters the transcriptome and therewith alters cancer- and bone-related biological properties of PC3 cells when seeded on the bone environment-mimicking Collagen 1 surface. The results of differential gene expression analysis and in vitro experiments implicate that SFRP2 is one of the key elements leading to the adaptation of cancer cells in the metastatic bone microenvironment (Figure 7). Collagen 1-driven phenotypical change in PC3*^SFRP2^* cells towards a mesenchymal cell may help to understand the unclear mechanisms of osteomimicry. The great similarities between metastatic prostate cancer cells and osteoblasts support tumor cell survival in the bone microenvironment [53]. Deciphering such bone adaptation mechanisms of prostate cancer cells may result in the identification of novel therapeutic targets. The isolation of a specific, highly aggressive cancer cell subpopulations could contribute to the understanding of the entire metastatic pathway. In particular, the identification of molecular markers that cause EMT [58] and support osteomimicry of primary tumor cells may be a promising approach to interfere with early prostate cancer metastasis.

By analyzing significantly differentially expressed genes between COL1-PC3 and COL1-PC3*^SFPR2^* cells, we found the up-regulation of the *Osteonectin* (*SPARC*) gene, which encodes an abundant non-collagenous protein of bone ECM [59] in COL1-PC3*^SFPR2^* cells. Prostate cancer cells are known to secrete a high level of SPARC protein when they are in the dormant stage of early bone metastasis [60]. The secretion of additional bone-related proteins in the bone microenvironment may allow cancer cells to settle, adapt, and survive more easily during metastatic progression. In accordance, COL1-PC3*^SFPR2^* cells had a high expression of *Pleiotrophin* (*Osteoblast Specific Factor-1*, *PTN*), which is considered as a poor prognosis marker in prostate cancer patients with bone metastasis [61], but it is generally known as a secreted growth factor produced by osteoblasts during early osteogenic differentiation and angiogenesis [62].

For the survival of the cells during metastasis extravasation and colonization, the cancer cells must regain their adhesion properties to the ECM after leaving the circulation, and the secretion of specific adhesion molecules could guide cancer cells towards the target site. Therefore, upregulation of adhesion-related genes might be a possible indication of osteotropism, which may be triggered by the demethylation of the *SFRP2* gene. Increased expression of *TNXB* (*Tenascin XB*) and *VWF* (*Von Willebrand Factor*) are both accepted as promising bone metastasis-related markers [63] (Figure 2A′, Appendix A) It is known that *TNXB* deficiency also causes significant bone loss in mice due to increase in osteoclasts [41], thus, the upregulation of *TNXB* triggered by SFRP2 overexpression in our in vitro study may also lead to reduced osteoclastogenesis in vivo, therefore further increasing bone formation (Appendix A). Furthermore, it was previously shown that *MMP11* upregulation is correlated to a poor prognosis of prostate cancer, and it is highly expressed in bone metastasis [64]. Furthermore, *Osteopontin* (*OPN*), a crucial ECM protein in the bone metastasis process [65], is a substrate of MMP17 [66]. Taken together, the increased expression of genes that are associated with bone formation in COL1-PC3*^SFRP2^* underlines osteomimicry in early and late metastatic stages.

While several studies suggest a role of SFRP2 in WNT signaling, our gene expression data showed that SFRP2 overexpression did not cause a reduction of WNT signaling, neither on CCP nor COL1-coated surfaces, yet it increases WNT related ligands expression as *WNT4* and *WNT10B*.

As our data indicate that SFRP2 causes a molecular and phenotypical switch of PC3 cancer cells towards more osteoblast-like cells, these novel findings may help to monitor the osteotropic activity of metastatic prostate cancer cells in all phases of bone metastasis. After cancer cells enter the circulation to metastasize, osteotropic activity of prostate cancer cells could be detected by elevated serum SFRP2 levels in patients, similar to breast cancer patients with a poor prognosis [23]. Therefore, the regular analysis of SFRP2 serum levels after the first diagnosis of prostate cancer can be a solution to identify potential metastasizes in an early stage. However, further experiments will be needed to demonstrate demethylation of *SFRP2* in primary and metastatic cancer tissues and to confirm elevated SFRP2 serum levels to detect the type and location of the early bone metastasis of prostate cancers. Furthermore, besides SFRP2 function to upregulate osteoblast-related gene expression, its effect on osteogenic differentiation must be verified by in vitro mesenchymal stem cell differentiation assays to understand the influence of SFRP2 in late bone metastasis.

## Figures and Tables

**Figure 1 cells-11-04081-f001:**
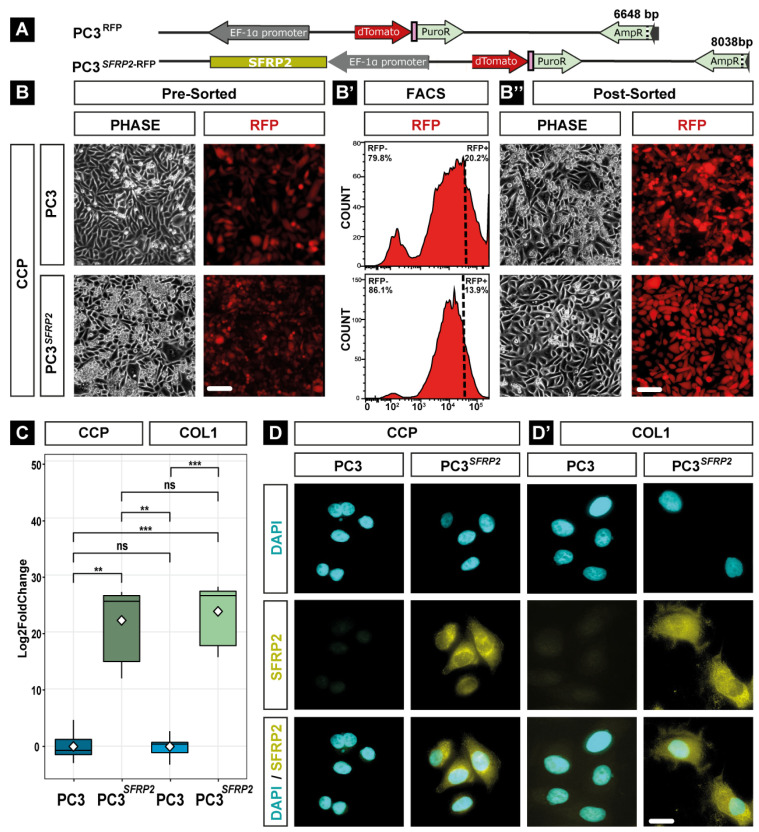
Generation and analysis of PC3 cells stably overexpressing SFRP2. The bicistronic expression plasmid contained either the coding sequences of *RFP* and puromycin for control cells (PC3*^RFP^*) or an additional coding sequence for human *SFRP2* (PC3*^SFRP2; RFP^*, **A**). Stably transfected cells were selected with puromycin for 14 days (**B**) and then enriched for high RFP fluorescence by FACS (**B′**), resulting in homogenous bright PC3 cell populations (**B″**). qPCR validated a significantly higher *SFRP2* expression in PC3*^SFRP2^*, when compared to PC3 cells, which was not affected by surface conditions (**C**). In addition, ICC against SFRP2 shows high expression of SFRP2 in the whole cytoplasm of PC3*^SFRP2^* cells compared to the controls, independent of the surface (**D**,**D′**). Significance level **: *p* ≤ 0.001, ***: *p* ≤ 0.001, ns: not significant. Scale bars: B/B″: 100 µm, D/D′: 10 µm.

**Figure 2 cells-11-04081-f002:**
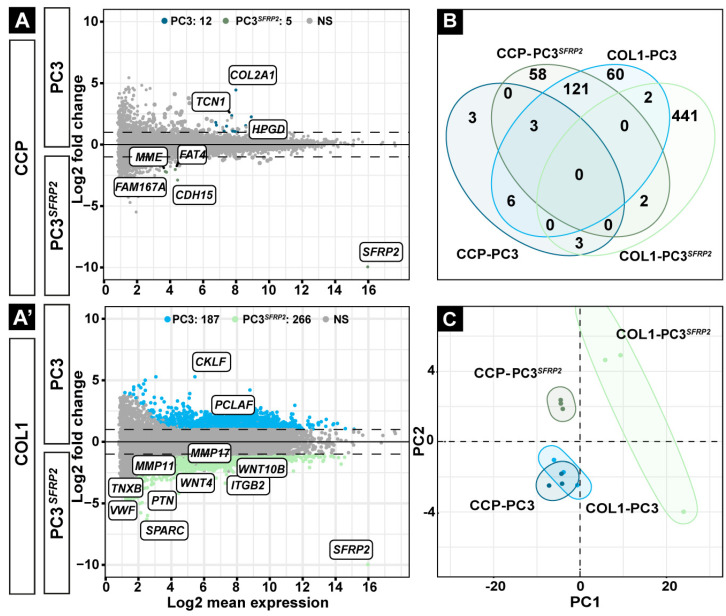
Differentially expressed gene (DEG) analysis of PC3 and PC3*^SFRP2^* cells. Next-generation RNA sequencing was performed to uncover genes that are affected by SFRP2 overexpression and surface condition. MA plots show gene expression differences between PC3 and PC3*^SFRP2^* on CCP (**A**) and COL1 (**A′**) surfaces. Genes that may influence osteomimicry were labelled in MA-plots. NS: not significant, blue: significantly upregulated in PC3, green: significantly upregulated in PC3*^SFRP2^*. The Venn diagram illustrates the number of unique upregulated genes specific to each group (**B**). Principal component analysis (PCA) revealed a strong transcriptional response of PC3*^SFRP2^* cells on COL1, when compared to PC3*^SFRP2^* on CCP surface (**C**).

**Figure 3 cells-11-04081-f003:**
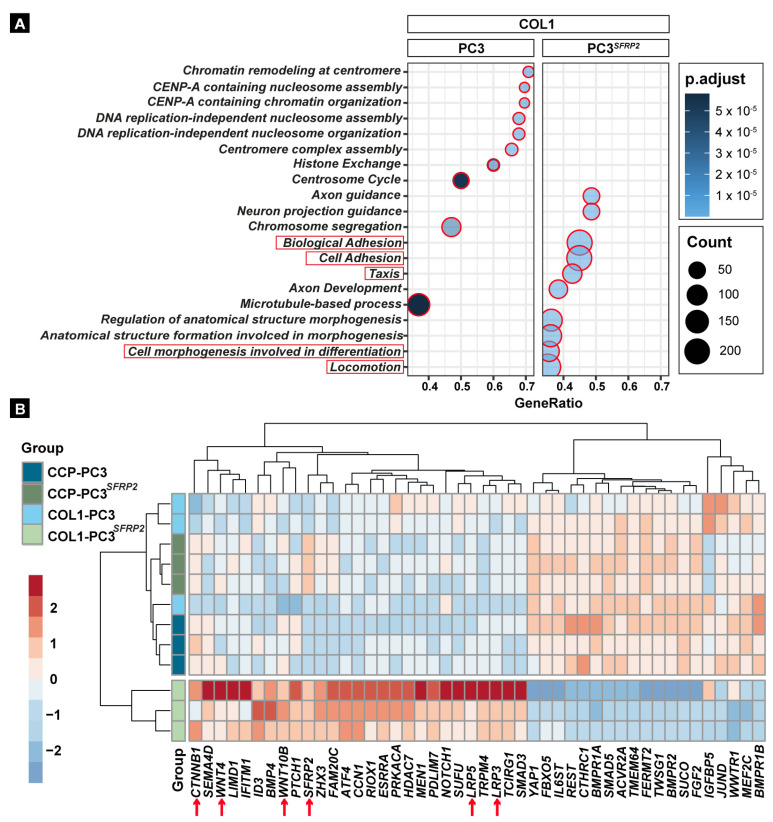
SFRP2 overexpression leads to osteotropic properties in PC3 cells on a COL1 surface. GO biological processes related to adhesion, taxis, and differentiation were significantly increased by SFRP2 overexpression compared to native PC3 cells when the cells were seeded on COL1 (**A**). Heatmap of osteoblast differentiation-related genes (GO:0001649) shows clear clustering of COL1-PC3*^SFRP2^* from all other groups (**B**). Arrows indicate WNT signaling-related gene expression in osteoblast differentiation.

**Figure 4 cells-11-04081-f004:**
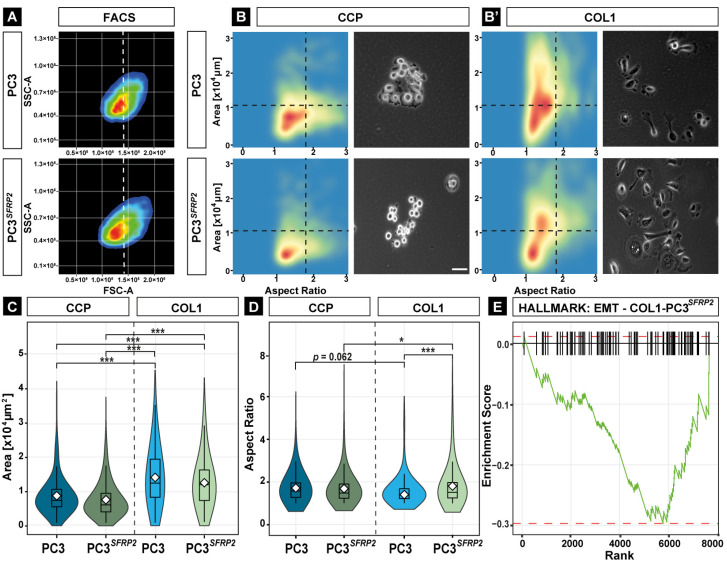
SFRP2 overexpression induces COL1-dependent EMT in PC3 cells. No morphological differences between PC3 and PC3*^SFRP2^* were observed in FACS analysis when cells were in suspension (**A**). On the CCP surface, PC3 cells are epithelial-like, while PC3*^SFRP2^* cells are more roundish (**B**). The COL1 surface induces an increase in cell size and elongation in both PC3 and PC3*^SFRP2^* (**B′**). Quantification of the cell morphology reveals that COL1 significantly increases cell size in PC3*^SFRP2^* cells (**C**) and elongation in COL1-PC3*^SFRP2^* cells (**D**). RNAseq analysis revealed that COL1-PC3*^SFRP2^* exhibited significant upregulation of the EMT pathway-related hallmark gene set when compared to CCP-PC3*^SFRP2^*. COL1-PC3*^SFRP2^* enrichment score < 0, CCP-PC3*^SFRP2^* enrichment score > 0 (**E**). * *p* ≤ 0.05; *** *p* ≤ 0.001. Scale bar: 200 µm.

**Figure 5 cells-11-04081-f005:**
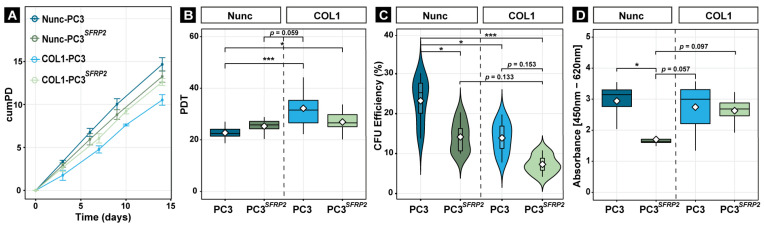
SFRP2 overexpression suppresses the reduced proliferation of PC3 cells on the COL1 surface. Cumulative population doubling (*cumPD*) (**A**) and population doubling time (*PDT*) (**B**) revealed that culturing PC3 cells on a COL1-coated surface reduced their proliferation in comparison to PC3*^SFRP2^*. COL1, and SFRP2 mutually significantly reduced CFU efficiency of PC3 cells (**C**). Metabolic activity is significantly reduced in PC3*^SFRP2^* cells on the CCP surface compared to native PC3. However, this significant difference is no longer observed when the cells are cultured on COL1 surface (**D**). Significance levels: * equals *p* ≤ 0.05; *** equals *p* ≤ 0.001.

**Figure 6 cells-11-04081-f006:**
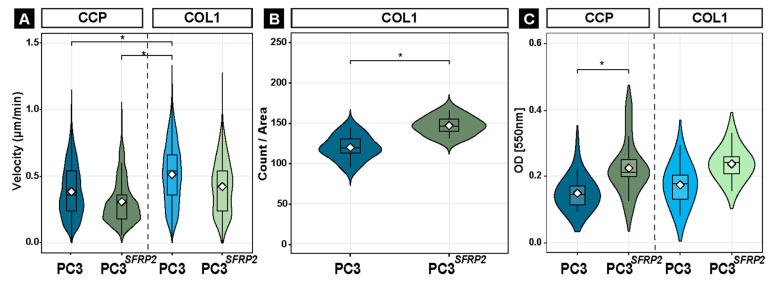
SFRP2 overexpression increases the osteotropic-like properties of PC3 cells on the COL1 surface. COL1-coating increases the random migration velocity of PC3 cells significantly, whereas overexpression of SFRP2 did not lead to such a significant increase in motility (**A**). Cellular invasion through a COL1 matrix shows that *SFRP2* overexpression significantly increases the invasion of the PC3 cells (**B**). Cell attachment assay shows an increase in attachment with *SFRP2* overexpression in PC3 cells cultured on both surfaces yet increase on COL1 is not significant. (**C**). Significance level: * equals *p* ≤ 0.05.

**Figure 7 cells-11-04081-f007:**
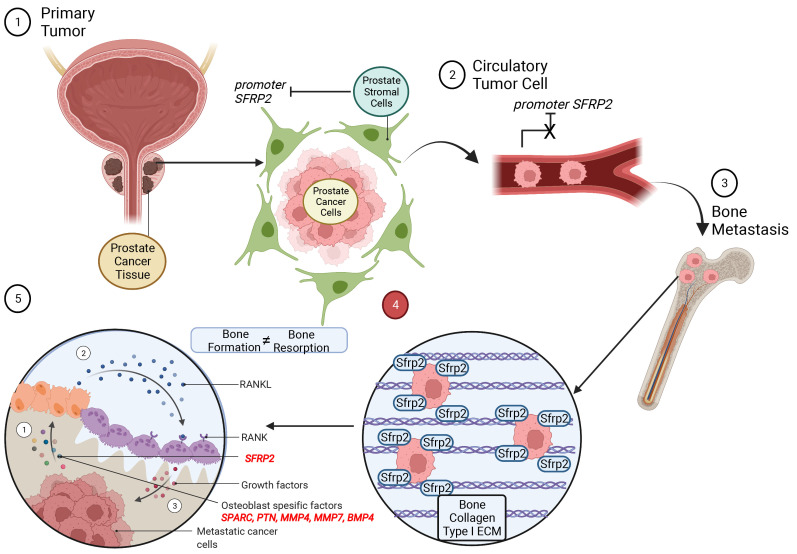
Visual representation of the potential effect of SFRP2 on prostate cancer cells during bone metastasis. After prostate cancer cells enter the circulation, SFRP2 expression leads them to attach and invade bone. PC3*^SFRP2^* cells start to secrete osteoblast-specific factors for osteomimicry and disturb the balance between bone formation and resorption. Created with BioRender.com (accessed on 14 June 2022, adapted from “Metastasis to Bone Disrupts Bone Homeostasis”).

## Data Availability

Gene expression data are deposited at https://www.ncbi.nlm.nih.gov/geo/query/acc.cgi?acc=GSE217979, accessed on 18 November 2022. All in vitro data are available from the corresponding author upon reasonable request.

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
