# Peer review of "SFRP2 Overexpression Induces an Osteoblast-like Phenotype in Prostate Cancer Cells"

_cells, 2022, doi:10.3390/cells11244081_

Round 1
Reviewer 1 Report
In the submitted manuscript authors demonstrated the role of SFRP2 overexpression in mediating the prostate cancer bone metastasis. Authors showed that SFRP2 can induce epithelial-mesenchymal transition and regulate the expression of the genes involved in osteoblast like phenotype. The introduction of the manuscript is well-directed and sufficiently detailed. Methods are mostly well adopted and logically performed except few places. However, the manuscript lacks some fundamental clarity and explanations:
The authors used collagen 1 coated flasks to mimic the bone microenvironment to show the response of SFRP2 overexpression in PC3 cells. The authors showed that SFRP2 overexpression did not influence the differential expression of genes when PC3 cells grew on CCP surface, which reflects the native tumor behavior. While on Col1 coated dishes, which mimic the bone microenvironment, significant overexpression of the genes was noted. These overexpressed genes showed EMT phenotype in PC3 cells in their GO analysis. However, it not clear why the PCa cells would show EMT at the metastatic site (COL1 coated flasks) and not in the primary tumor (CCP coated flasks). It is logical to assume that PCa cells would have more EMT like phenotype at the primary tumor and MET at that metastatic site.
Authors mentioned on pages 8 and 9 that comparison of PC3 and PC3SFRP2 cells cultured on CCP surface showed 17 genes to be differentially expressed and differential expression of 453 genes on COL1 surface; however comparison of Venn diagram in figure 2B and supplementary figure S1C (showing up and downregulated genes) did not match these numbers.
The unique upregulated and downregulated genes in figure 2B, supplementary figure S1C and the result in section 3.3 is not clear. How the up or downregulation was calculated? For example, in results 3.3 authors mentioned that they analyzed 453 DEGs on the COL1 surface due to SFRP2 overexpression; one would assume that these 453 DFGs are either up or down-regulated in one condition (PC3 SFRP2) compared to other (COL1-PC3), if that is true how authors considered 20 significantly upregulated biological process in COL1-PC3 and COL1-PC3SFRP2. Is it not like, upregulated proteins in COL1-PC3 means downregulated in COL1-PC3SFRP2, since up and downregulation is relative to these two conditions (COL1-PC3 v/s COL1-PC3SFRP2), then how authors showed two distinct GO pathways in figure 3a?
Reviewer 2 Report
In this study, Olken et al. investigate the role of SFRP2 in regulating the metastatic potential of PC3 cells. They establish cell lines that stably over-express SFRP2 and conduct RNA-seq analysis on cells plated on plastic vs. those on surfaces coated with COL1. Whereas few targets were differentially expressed when cells were cultured on CCP, SFRP2 induced differential expression of 453 genes when plated on COL1. Notably, WNT target genes were absent on this list, although WNT4A and WNT10B transcripts were induced. Further analysis of these genes pointed to processes associated with EMT and cellular metastasis. In agreement with these observations, the authors found that both PC3 and PC3SRFP2cells displayed an elongated morphology on COL1-coated surfaces and that PC3SRFP2cells expressed EMT genes in comparison to PC3SRFP2plated on CCP. SFRP2 expression overcame reduced proliferation phenotypes of PC3 cells when cultured on COL1. Finally, SFRP2 expression increased invasive properties of PC3 cells through COL1.
This was a well-written and thorough study with interpretations supported by presented results. The authors acknowledge that more work is needed to determine when theSFRP2promoter is methylated during the carcinogenic/ metastasis process and to identify the source of circulating SFRP2 in the metastatic niche. This study sheds significant light on the role of SRFP2 in osteomimicry in prostate cancer and provides the foundation for future work investigating the mechanistic underpinnings of SRFP2-dependent signaling in AdPCA.
Minor comments:
Line 100: “adopt” should be changed to adapt.
Line 379: “Beside” should be besides.
Round 2
Reviewer 1 Report
The author's response to the reviewer's comment is satisfactory and the manuscript can be accepted in the current for.